# The Impact of the Nephrotoxin Ochratoxin A on Human Renal Cells Studied by a Novel Co-Culture Model Is Influenced by the Presence of Fibroblasts

**DOI:** 10.3390/toxins13030219

**Published:** 2021-03-18

**Authors:** Gerald Schwerdt, Michael Kopf, Michael Gekle

**Affiliations:** Julius-Bernstein-Institut für Physiologie, 06112 Halle, Germany; michael.kopf@uk-halle.de (M.K.); michael.gekle@medizin.uni-halle.de (M.G.)

**Keywords:** ochratoxin A, cell culture, energy metabolism, apoptosis-necrosis balance, mitochondria

## Abstract

The kidney is threatened by a lot of potentially toxic substances. To study the influence of the nephrotoxin ochratoxin A (OTA) we established a cell co-culture model consisting of human renal proximal tubule cells and fibroblasts. We studied the effect of OTA on cell survival, the expression of genes and/or proteins related to cell death, extracellular matrix and energy homeostasis. OTA-induced necrosis was enhanced in both cell types in the presence of the respective other cell type, whereas OTA-induced apoptosis was independent therefrom. In fibroblasts, but not in tubule cells, a co-culture effect was visible concerning the expression of the cell-cycle-related protein p21. The expression of the epithelial-to-mesenchymal transition-indicating protein vimentin was independent from the culture-condition. The expression of the OTA-induced lncRNA WISP1-AS1 was enhanced in co-culture. OTA exposure led to alterations in the expression of genes related to energy metabolism with a glucose-mobilizing effect and a reduced expression of mitochondrial proteins. Together we demonstrate that the reaction of cells can be different in the presence of cells which naturally are close-by, thus enabling a cellular cross-talk. Therefore, to evaluate the toxicity of a substance, it would be an advantage to consider the use of co-cultures instead of mono-cultures.

## 1. Introduction

Due to its excretory function, the kidney is threatened by a variety of harmful substances such as drugs or food contaminants, like mycotoxins leading to acute or—even worse—chronic kidney diseases with a prevalence of about 10% [1,2]. To understand the mechanisms of the nephrotoxic action it is helpful to find strategies to alleviate these detrimental scenarios and many studies have been performed to solve the question of why and how kidneys are endangered [3,4,5].

To study the influences of a substance on an organism, it is often difficult to use whole animals because of ethical concerns and organizational, costly and elaborate prerequisites. Furthermore, a transfer of knowledge to the human situation is associated with uncertainties. Therefore, cell culture models have been established and are used widely, and have the advantage that a specific cell type and its response to a substance or to a treatment can be studied under controlled conditions. Although many and important findings have been made using this approach, some disadvantages are inherent: cells of a cell line often have been immortalized by mutagenesis or other—sometimes drastic—methods [6]. This allows easy handling and long usage but with the hazard that results found in a specific model system may not be transferable to the situation in the whole organ or organism. To overcome this disadvantage, instead of immortalized cell lines, primary cells can be used, but the generation of primary cells is often very difficult and requires advanced technical skills. Additionally, primary cells often do not survive for a long period of time and need special individual culture conditions. But primary cells are a step closer to natural conditions and at least sometimes it turned out that they are more sensitive to, e.g., toxic stimuli as cell lines are [7], meaning that cell lines might be more robust. Another disadvantage is the fact that cells are often kept in monoculture, i.e., without the influences of other cell types, which in their home organ usually are close-by. Therefore, it might be a step towards a more realistic situation to study the response of a cell type to a substance or treatment in the presence of those cells, which are in the native organ in close proximity.

In the kidney, proximal tubule cells are surrounded by fibroblasts and a—probably mutual—influence can be assumed. This is also the case in kidney damaging scenarios that in most cases lead to tubulo-interstitial inflammation and fibrosis [8,9], and are decisive for the decline of kidney function. Because of their transport and enzymatic capabilities, renal proximal tubule cells are endangered by a variety of potential toxic substances such as, e.g., drugs or their remnants or mycotoxins. One role of the surrounding fibroblasts is to furnish the extracellular matrix by release of collagens and other matrix components and therefore to participate in the integrity of the tissue [10]. But they are also—together with epithelial cells—involved in inflammatory processes or in fibrotic kidney diseases [11] with the risk of developing renal failure.

An intensively studied mycotoxin with relevance for human health is ochratoxin A (OTA) [12,13]. It can be found in a variety of foodstuffs [12,14] and to avoid its exposure and uptake is almost impossible [9,15]. This leads to the observation that OTA is detected frequently in human blood in low nanomolar concentrations [16]. In exposed animals, OTA leads to kidney failure and fibrotic changes [17,18]. OTA exposure is assumed to be involved in human kidney diseases [19]. In human primary proximal tubule cells, a toxic effect of OTA has been shown, which is also observable in human primary fibroblasts, although it is not as prominent as in proximal tubule cells [7]. The mechanisms behind the toxic action of OTA are still not completely understood and are subject of many ongoing studies. How far neighboring cells with different functions interfere and thereby modulate cell function is almost not known but is expectable. In a previous study using a co-culture model consisting of rat kidney proximal tubule and fibroblast cells, it turned out that a kind of crosstalk between both cell types takes place, leading to the observation that effects of OTA as epithelial-to-mesenchymal transition (EMT) occurred only under co-culture conditions [20]. Another conclusion drawable from that study and others was that rat cells are more robust concerning the tolerance to OTA as compared to human proximal tubule cells [7,20] and therefore a model system based on human cells is required to closer evaluate the human situation and the risk of OTA exposure.

Therefore, in the present study, we establish an advanced cell co-culture model consisting of human proximal tubule cells (HK2 cells) and human fibroblasts (CCD-1092SK cells) to study the effects of OTA on cell survival (apoptosis, necrosis) and expression of some exemplarily chosen genes related to cell cycle, cell death, extracellular matrix, and metabolism. Similar to previous studies using rat cells [20], the human proximal tubule cells were placed on filter devices and the filters were put above a layer of fibroblasts seeded on the bottom of a petri dish so that the basolateral side of the epithelial cells faces towards the fibroblasts, enabling a kind of conversation between both cell types.

## 2. Results

### 2.1. Protein, Lactate Dehydrogenase Release and Caspase-3 Activity

To obtain a first impression about possible effects of culture conditions itself as well as about effects on OTA-induced alterations, we compared caspase-3 activity as a measure for apoptosis of cells grown in monoculture with the activity of cells grown in co-culture incubated with or without 100 nM OTA for two points of time, 24 and 48 h. In addition, lactate dehydrogenase (LDH) release as a measure for necrosis was determined as well as protein content to give an overall impression on cell status. Therefore, equal amounts of cells were placed either in the well bottom (fibroblasts) or onto a filter (proximal tubule cells). After reaching confluence, filters were placed into the wells in which the fibroblasts were located (see graphical abstract). As shown in Figure 1A, almost no culture-condition-dependent effects on protein content could be observed in both cell types after 24 or 48 h (see also Appendix A). In fibroblasts, OTA exposure led to a small increase in protein content whereas in tubule cells OTA led to a slight decrease of protein content showing that OTA might have a negative effect on tubule cells. These effects were almost independent from the culture condition in both cell types.

To further explain the effect on protein content, we studied apoptosis and necrosis. Compared to fibroblasts, OTA had a clear effect on apoptosis in tubule cells after 24 h exposure with about 2.5-fold increase in activity (Figure 1B). After 48 h exposure the increase was still observable but not as distinctive as after 24 h. However, these increases were almost independent from culture conditions except that after 48 h in the presence of fibroblasts the caspase activity in the tubule cells was slightly reduced, indicating a modest protecting effect of the co-culture. However, a protecting effect of co-culture was observable for the fibroblasts, especially in the presence of OTA.

In Co-culture, LDH release was clearly enhanced in fibroblasts and tubule cells when compared to monoculture conditions independent of the presence of OTA (Figure 1C). In tubule cells, OTA led to an increase of LDH released into the media especially after 48 h exposure, which was not as pronounced in fibroblasts.

Taken together, the presence of the respective other cell type led to enhanced necrosis but to less apoptosis so that the overall protein content was not remarkably changed. The effects of OTA on apoptosis and necrosis were also mostly independent from the presence or absence of the other cell type.

### 2.2. Western Blot and mRNA Expression

#### 2.2.1. CDKN1A/p21

Cell cycle was shown to be influenced by OTA and it could be shown that the p21 protein which is involved in cell cycle was upregulated by OTA in tubule cells [21]. To investigate how far the protein, as well as the expression of mRNA coding for p21, is influenced by the presence of fibroblasts, we performed Western blots and RT-PCR. As shown in Figure 2A,B and Figure 3, 48 h exposure to 100 nM OTA led to an increase of p21 protein amount in mono but also in co-culture conditions in tubule cells. In fibroblasts under co-culture conditions, OTA had no effect on p21-protein expression although the mRNA expression was increased by OTA independent from the presence of the other cell type. Interestingly, under co-culture conditions, OTA exposure did not further increase p21 protein expression. In tubule cells, the mRNA expression was not altered by the presence of fibroblasts but in fibroblasts in co-culture the p21 mRNA expression was enhanced not only in OTA-exposed but also already in cells not exposed to OTA (see also Appendix A). This shows that the presence of the other cell type has an influence on p21 protein expression, especially in fibroblasts.

#### 2.2.2. Cyclooxygenase 2 (COX2)

It has been shown that cyclooxygenase 2 (*COX2*) protein as well as mRNA levels are increased during kidney failure [22]. As shown in Figure 2C,D and Figure 3, only in fibroblasts the protein expression was increased by OTA exposure. In addition, in the presence of tubule cells, COX2 protein expression was enhanced in untreated as well as in OTA-exposed cells, demonstrating a clear influence of tubule cells. The mRNA expression, however, was not influenced by OTA and a very slight effect of co-culture occurred by OTA exposure. In contrast, in tubule cells, the mRNA and protein expression of COX2 was completely independent from OTA or the presence of fibroblasts. That shows that concerning COX2 tubule cells can influence fibroblasts but fibroblasts have no influence on tubule cells.

#### 2.2.3. Fibronectin

Kidney failure is often accompanied by fibrosis. During fibrosis, an accumulation of extracellular matrix takes place and one observation besides other is an increase of fibronectin protein amount [23]. Therefore, the OTA-dependent alteration of fibronectin-coding mRNA and protein expression was determined in mono- or co-culture conditions. As shown in Figure 3 and Figure 4A,B, 48 h exposure of tubule cells to 100 nM OTA led to a decrease of intracellular fibronectin both in mono-and co-culture. The OTA-effect was lower in co-culture. Moreover, the amount of mRNA coding for fibronectin was reduced by OTA exposure independently from the culture conditions and the presence of fibroblasts led to lower mRNA expression in control and OTA-exposed cells. In contrast, in fibroblasts the fibronectin expression was almost not altered neither by OTA nor by culture conditions because of a great variability, especially in the Western blots. A tendency towards OTA-induced expression might be visible in mono-culture.

#### 2.2.4. Vimentin

Vimentin is a protein whose abundance increases when epithelial-to-mesenchymal transition (EMT) takes place and EMT development can lead to kidney failure [23]. Therefore, the OTA-dependent alteration of vimentin mRNA and protein expression was determined in mono- and co-culture conditions. As shown in Figure 3 and Figure 4C,D, 48 h exposure of tubule cells to 100 nM OTA led to a lower abundance of vimentin protein both under mono- and co-culture conditions. However, in the presence of fibroblasts the vimentin protein expression was independent from culture conditions. The expression of mRNA coding for vimentin was almost not altered neither by OTA nor by culture conditions with the exception that OTA exposure in co-culture showed a slight increase. In fibroblasts, the vimentin protein expression was completely independent from OTA exposure as well as from the presence of the tubule cells. In addition, vimentin-coding mRNA expression was almost not altered.

### 2.3. Expression of Some Selected Genes

To further test exemplarily in how far the culture conditions may affect also the expression of other RNAs, we selected some genes, which play a role in apoptosis-necrosis, cancer development, or energy metabolism (see also Appendix A).

#### 2.3.1. WISP1-AS1

*WISP1-AS1* is a long non-coding RNA (lncRNA) induced by OTA affecting transcriptional regulation and playing a role in the apoptosis-necrosis balance and probably in cancer development [24]. As seen in Figure 5A, 48 h exposure to 100 nM OTA led in both cell types to a marked increase in the expression of that lncRNA. Without OTA, the presence of the respective other cell type had almost no influence on the expression. However, in the presence of OTA, its expression was higher in co-culture as compared to mono-culture.

#### 2.3.2. GDF15

Growth differentiation factor 15 (*GDF15*) is a member of the transforming growth factor superfamily responding to stress. It is discussed as a biomarker also for kidney diseases or as a predictor for survival of kidney transplant patients [25,26]. The expression of mRNA coding for *GDF15* was enhanced in both cell types after OTA exposure as shown in Figure 5B. This OTA-induced effect was favored in fibroblasts when tubule cells were in the vicinity. In tubule cells, however, the expression was independent of the presence of fibroblasts (Figure 5B).

#### 2.3.3. CDK2

Cyclin-dependent kinase 2 *(CDK2)* was identified by weighted correlation network analysis as a major regulator of OTA-induced cell cycle dysregulation [21]. In fibroblasts in monoculture the expression of the mRNA coding for *CDK2* was not altered by OTA. Furthermore, the presence of tubule cells did not affect the mRNA expression. However, in tubule cells, OTA led to a slightly enhanced expression only in the presence of fibroblasts (Figure 5C).

#### 2.3.4. Glycogen and Glucose-Related Proteins: *PYGM, GYS1* and *GLUT1 (SLC2A1)*

The kidney is also involved in glucose homeostasis and can provide the body with glucose either by gluconeogenesis or by mobilizing glycogen stores [27]. Glycogen phosphorylase (*PYGM*) plays a role in the decomposition of glycogen stores, thereby mobilizing glucose [28]. As seen in Figure 6A, 48 h exposure to 100 nM OTA led to a marked increase in the expression of the mRNA coding for glycogen phosphorylase, especially in the tubule cells. However, in tubule cells this increase was not dependent on culture condition whereas in fibroblasts the OTA-induced expression was higher in the presence of tubule cells as compared to the condition without tubule cells. Glycogen synthase 1 (*GYS1*) catalyzes the opposite reaction and whereas the expression of the phosphorylase was upregulated, the expression of the synthase was down regulated by OTA in tubule cells and in fibroblasts in co-culture (Figure 6B). This indicates a glucose-mobilizing effect of OTA. Interestingly, the mRNA expression of the glucose transporter GLUT1 *(SLC2A1)* was upregulated by OTA, too (Figure 6C), underlining the idea of an enhanced glucose demand due to OTA exposure maybe due to impaired mitochondria.

#### 2.3.5. Mitochondria-Related Proteins: *NDUFB10* and *MRPS16*

There are indications that OTA exposure can lead to a decrease of the mitochondrial potential in kidney cells [24] showing that mitochondrial function may be influenced by OTA exposure, which might be additionally influenced by the presence of fibroblasts. Therefore, representative of other RNA coding for mitochondrial proteins, we show here the expression of the mRNAs coding for *NDUFB10* and *MRPS16*. *NDUFB10* codes for the mitochondrial NADH:ubiquinone oxidoreductase subunit B10, which is a part of the mitochondrial respiratory complex I and highly expressed in heart and kidney (NCBI gene. Available online: https://www.ncbi.nlm.nih.gov/gene/4716 (accessed on 17 March 2021)). The *MRPS16* gene codes for the mitochondrial ribosomal protein S16, which plays a role in mitochondrial protein synthesis. As shown in Figure 7, 24 h OTA exposure led in both cell types to a decreased expression of both mRNAs. In tubule cells, the decreased expression of *NDUFB10* gene was not influenced by the presence of fibroblasts, whereas in fibroblasts the OTA-induced reduction of *NDUFB10* expression was slightly rescued in the presence of tubule cells (Figure 7A). In tubule cells, the already lowered expression of the gene coding for the mitochondrial ribosomal subunit by OTA was additionally lower in the presence of fibroblasts, whereas in fibroblasts the presence of tubule cells led to a slightly higher expression of *MRPS16* mRNA (Figure 7B). This shows that mitochondrial function can be affected by OTA but that the OTA effect can additionally be modified when the two cell types are close together.

## 3. Discussion

The kidney is endangered by a variety of nephrotoxic substances with the risk of acute or chronic kidney failure [1]. For the study of the impact of nephrotoxic substances on different cell types and to reduce animal handling, cell cultures have been widely applied. These cell cultures have the additional advantage that experimental conditions can be controlled and specific effects of a substance investigated using a defined cell type. Besides these undisputed advantages, some considerations remain: for example, in the “home organ” one cell type (e.g., proximal tubule cells in the kidney) is surrounded by other cell types (e.g., fibroblasts) with manifold interdependencies. Therefore, the reaction to a substance observed when using only one cell type may not be the same as in the presence of the cells close-by in the original tissue. In a model consisting of two different rat renal cell types, it has been shown that a cross-talk exists between tubule and fibroblast cells, leading to reactions appearing solely when the two cell types were close-together [20]. However, when comparing the results of that study with findings observed using human renal tubule cells it emerges that rat cells obviously are more robust than the human cells concerning their reaction to treatment with a ubiquitous nephrotoxin, ochratoxin A (OTA) [7]. Therefore, a co-culture model consisting of human cells is necessary. We established such a model by using the human proximal tubule cell line HK2 and human fibroblasts. HK2 cells were grown on filter inserts and brought together with fibroblasts, grown on the bottom of a 6-well plate. After recording basic parameters as apoptosis and necrosis, we used this model to obtain initial data on the effect of ochratoxin A on the expression of some proteins and RNAs related to cell cycle, EMT, and cellular metabolism.

### 3.1. Cell Survival

Based on protein content, it seemed that neither OTA nor the culture-condition had a remarkable effect. However, the relationship between apoptosis and necrosis was shifted towards necrosis in both cell types when cells were cultured together. OTA is known to induce apoptosis in tubule cells [29] and to a lesser extend also in fibroblasts [7]. This is reflected also in the present results. Interestingly, the presence of the respective other cell type led to decreased apoptosis rates in both cell types but to enhanced LDH release. This is a first hint that already the presence of another cell type can influence cellular function and that the reaction of cells to a toxic substance can be different in co-culture compared to mono-culture.

### 3.2. Protein and RNA Expression

We extended our studies by the determination of protein and RNA expression of some exemplarily chosen proteins and the RNAs coding for them. As OTA was shown to influence cell cycle [21,30], the expression of CDKN1A/p21 was studied. According to the findings by Dubourg et.al [21], OTA led to an enhanced expression not only of *CDKN1A* mRNA but also of CDKN1A/p21 protein in tubule cells. In these cells, the presence of fibroblasts did not influence *CDKN1A* mRNA expression, whereas in fibroblasts a clear co-culture effect was visible with enhanced mRNA abundance only visible in co-culture. However, this increase in mRNA abundance was not completely mirrored by protein expression, suggesting further regulatory mechanisms. Cell cycle studies may be added to get a further insight into the impact on cell cycle. Cyclin-dependent kinase 2 *(CDK2)* was found to play a role in OTA-induced dysregulation of the cell cycle [21]. In tubule cells, its mRNA expression was enhanced by OTA in co-culture whereas fibroblasts were not influenced by tubule cells. Together, the results suggest that the cell cycle in tubule cells is influenced by the presence of fibroblasts.

Another co-culture effect was observed for the expression of *COX2* in fibroblasts. For fibroblasts the presence of tubule cells led to a clearly enhanced protein expression (similar to results observed in rat cells [20]), which was not visible for the tubule cells which did not show any culture-dependency. This indicates that the role of fibroblasts in inflammatory kidney diseases may have been underestimated by studies using fibroblasts in mono-culture.

The intermediate filament protein vimentin is constitutively expressed in fibroblasts and is increasingly expressed during epithelial-to-mesenchymal transition (EMT) in epithelial cells [10]. Although it has been shown that prolonged OTA exposure can lead to enhanced expression of collagen III or fibronectin in tubule cells [7], in the present study the expression of the EMT-indicating protein vimentin in tubule cells was even lower after OTA exposure, independent of culture conditions. In addition, the expression of fibronectin mRNA and protein was rather lower and not enhanced. Furthermore, in fibroblasts, no altered expression was demonstrable so that these findings do not argue in favor of an EMT induced by OTA, as shown by others [31].

To further test whether co culture can influence the cellular answer to stress induced by OTA, we determined the expression of some exemplarily chosen RNAs related to apoptosis-necrosis, cancer development and energy metabolism.

Long non-coding RNAs play a significant role in many cellular regulatory processes and their derailing, and also in renal fibrosis or cancer [32]. *WISP1-AS1* is a long non-coding RNA induced by OTA and expressed in renal tumor cells [24]. We found its upregulation by OTA in both cell types and co-culture enhanced the upregulation. This allows the aggravation of the effect of *WISP1-AS1* on the apoptosis-necrosis balance and probably tumor formation. The enhanced LDH release observed in co-culture can therefore at least partially be explained by the enhanced content of *WISP1-AS1*, which was shown to be necessary for OTA-induced necrosis [24].

*WISP1-AS1* was also suspected to play a role in metabolism including mitochondria [24]. In gastric epithelium cells, OTA was shown to cause mitochondrial dysfunction [33]. For kidney cells, there are controversial results as to whether OTA exposure has an influence on mitochondrial potential or not [21,24]. A reduced mitochondrial potential might be the result of impaired mitochondria or might proceed mitochondrial damage. The reduced expression of mitochondrial proteins may lead to impaired function mirrored or followed by altered mitochondrial potential. Impaired mitochondrial function forces the cell to use alternative pathways to assure energy supply. Energy supply under inappropriate mitochondrial contribution can be maintained by an increased use of glycolysis leading to enhanced production of lactate. However, in HEK293 cells, a human embryonic kidney cell line, it was found that OTA led rather to a reduction of glycolysis and the enhanced amount of lactate due to lactate production from glutamine was dependent on the expression of the lncRNA *WISP1-AS1* [24]. In contrast, in gastric epithelia cells, it could be shown that OTA exposure leads to a reprogramming of glucose metabolism towards glycolysis and less tricarboxyic acid cycle activity [34]. We can show here that (1) OTA leads to a reduced mRNA expression of two representatively chosen mitochondrial proteins, which play a role in mitochondrial protein synthesis (*MRPS16*) and energy production (*NDUFB10*) and (2) that glycogen-handling enzymes were regulated in a way that enhanced glucose can be mobilized (*GYS1* down and *PYGM* up). Additionally, a higher GLUT1 transport capacity seems to be induced. However, these OTA-induced alterations were almost independent from culture conditions.

In conclusion, we have shown that under co-culture conditions, the reaction of the cells can be different from the reactions observed in mono-culture, although not all parameters studied here were culture-dependent. However, based on the findings presented here, the use of co-culture should be preferred if possible, thus avoiding the possibility to oversee the effects not taking place when solely one cell type is studied. The question remains of how the cells communicate between each other. Studies in a rat co-culture model revealed a COX2-dependent mechanism [20] and in mice, retinoic acid seems to participate in cellular cross-talk of kidney cells [35]. Additionally, the question remains, if, why and how OTA interferes with cellular energy metabolism and the role of mitochondria therein.

## 4. Materials and Methods

### 4.1. Cell Culture

Human proximal tubule cells and fibroblasts were purchased from ATCC (Rockville, MD, USA; HK2: CRL-2190 and CCD-1092SK:CRL-2114). Both were cultured in DMEM-HamF12 media (PAN Biotech, Aidenbach, Germany) containing 10% fetal calf serum. Media were changed every week. 24 h prior to and during OTA exposure, cells were held in serum-free media. For co-culture experiments, HK2 cells were seeded onto a filter (Falcon, Corning GmbH, Wiesbaden, Germany, pore size 0.4 µm) immersed in media in a 6-well-plate whereas fibroblasts were seeded on the well bottom of another 6-well plate. After reaching confluence (usually three days after seeding) the filter with the proximal tubule cells were placed into the well with the fibroblasts in serum-free media. The media volume was 2 mL on the basolateral side of the tubule cells and 900 µL apical. In monoculture, HK2 cells were seeded onto filters, which were further handled as in co-culture except that no fibroblasts were present.

### 4.2. Determination of LDH and Caspase-3 Activities and of Protein Content

For lysis, cells were washed twice in ice-cold PBS buffer, collected and lysed in MOPS-Triton buffer (20 mM 3-(*N*-morpholino)propanesulfonic acid, pH 7.4, 0.1% Triton X100). Protein content in cell lysates was determined using bicinchoninic acid [36,37]. LDH activity in media or cell lysates as a measure for necrosis was determined according to Bergmeyer [38] as described in detail in [20]. Caspase-3 activity as a measure for apoptosis was determined using the florigenic caspase-3 substrate (DEVD-AFC) as described in [39]. Briefly, 60 µL cell lysate was incubated with 65 µL reaction buffer (20 mmol/L piperazine-1,4-bis(2ethanesulfonic acid (PIPES), 4 mmol/L EDTA, 0.2% 3-[(3-cholaminopropyl)dimethylammonio]-1-propanesulfate (CHAPS), 10 mmol/L dithiotreitol (DTT), pH 7.4) containing 42 µmol/L DEVD-AFC (Asp-glu-val-asp-7-amino-4-trifluoromethylcoumarin, end-concentration) at 37 °C. Fluorescence of the cleaved product (AFC) was measured at 400 nm excitation and 505 nm emission. Cleaved AFC was quantified by a calibration curve using known AFC concentrations.

### 4.3. RT-PCR

Isolation of total ribonucleic acid (RNA) was performed using Trizol reagent (Life Technologies, Darmstadt, Germany). Cells were washed and thereafter lysed with Trizol reagent and transferred into a reaction tube. After addition of chloroform and centrifugation (12,000× *g*), the upper phase was collected and mixed with ice-cold isopropanol. After centrifugation (12,000× *g*) the supernatant was removed and the pellet washed twice with 75% ethanol and finally solved in water. Reverse transcription was performed using a commercial kit from Invitrogen (Thermo Fisher Scientific, Waltham, MA, USA) according to their instructions. Real-time PCR was performed using a SYBR Green reagent (Invitrogen). Primers were synthesized by Microsynth AG, Balgach, Switzerland. Primer sequences are shown in Table 1. Fold change of gene expression was calculated by the 2ΔΔCt method using the expression of *EEF2* and *RPS17* as references. The expression of these two genes turned out to be the less altered ones (if at all) in RNA sequencing data when comparing OTA treated with non-treated HK2 cells (non-published results).

### 4.4. Western Blots

After separation of the proteins by sodiumdodecylsulfate-polyacrylamide gel electrophoresis (SDS-PAGE), proteins were transferred onto a nitrocellulose membrane. Thereafter, free binding sites of the membrane were blocked by a 5% solution of non-fat dry milk in TRIS-buffered saline (3 mM TRIS base, 140 mM NaCl, 0.17 mM Tris-HCl, pH 7.4) containing 0.1% Tween20. The first antibodies diluted in TRIS saline + 5% bovine serum albumin (TRIS-BSA, for dilutions see Table 2) were added and membranes incubated overnight. After washing, fluorescence-coupled secondary antibodies in TRIS-BSA were added for 90 min. Fluorescence of the second antibodies was recorded using a LICOR detection system.

### 4.5. Statistics

The significance of difference was determined by the unpaired Student’s *t*-test. *p* ≤ 0.05 was considered to be statistically significant and indicated by an * in the figures.

## Figures and Tables

**Figure 1 toxins-13-00219-f001:**
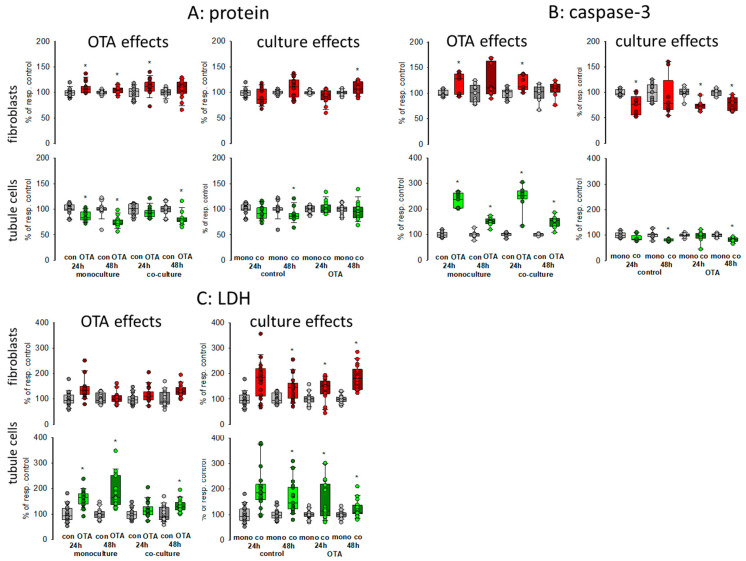
Effects of ochratoxin A (OTA) and/or culture conditions on protein content (**A**), caspase-3 activity (**B**), and lactate dehydrogensase (LDH) release (**C**). Cells were cultivated either in mono- or in co-culture and exposed to 100 nM OTA for 24 or 48 h. *n* = 3–6, *n* = 14–18 (protein, LDH) or 8–9 (caspase-3). * indicates a *p* < 0.05 to non-OTA-exposed cells (comparing the OTA effects, resp. left side) or to cells in mono-culture (comparing culture effects, resp. right side).

**Figure 2 toxins-13-00219-f002:**
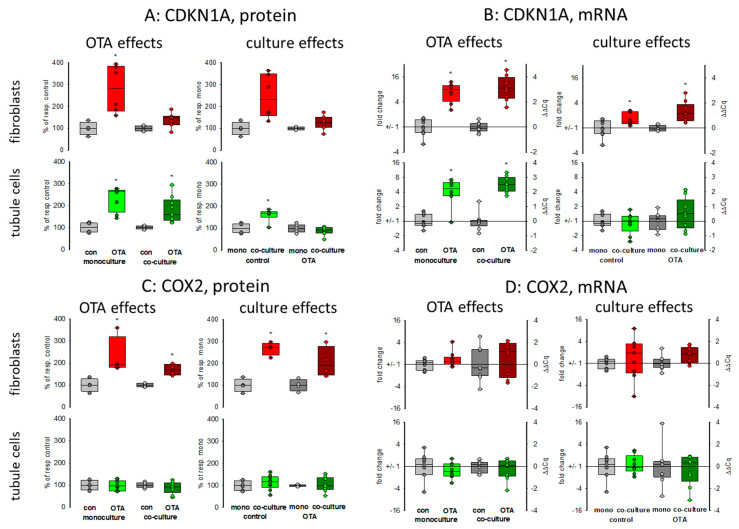
Effects of OTA and/or culture conditions on protein and mRNA expression of CDKN1A/p21 (**A**,**B**) and cyclooxygenase 2 (COX2) (**C**,**D**). Cells were cultivated either in mono- or in co-culture and exposed to 100 nM OTA for or 48 h. *n* = 3, *n* = 4–9 (protein and 8–9 for mRNAs. * indicates a *p* < 0.05 to non-OTA-exposed cells (comparing the OTA effects, resp. left side) or to cells in mono-culture (comparing culture effects, resp. right side).

**Figure 3 toxins-13-00219-f003:**
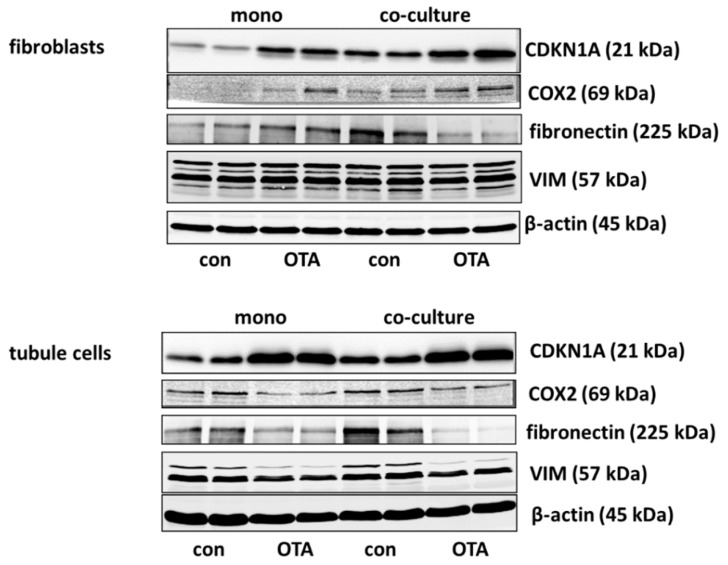
Supplementary representative Western blots to Figure 2A,C and Figure 4A,C.

**Figure 4 toxins-13-00219-f004:**
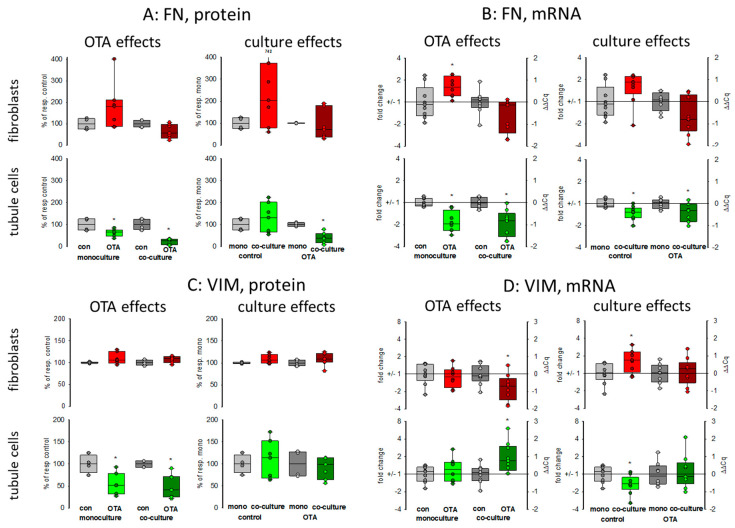
Effects of OTA and/or culture conditions on protein and mRNA expression of fibronectin (*FN*) (**A**,**B**) and vimentin (*VIM*) (**C**,**D**). Cells were cultivated either in mono- or in co-culture and exposed to 100 nM OTA for 48 h. *n* = 3; *n* = 4–8 (FN protein), 4–7 (VIM protein) and 8–9 for mRNAs. * indicates a *p* < 0.05 to non-OTA-exposed cells (comparing the OTA effects, resp. left side) or to cells in mono-culture (comparing culture effects, resp. right side).

**Figure 5 toxins-13-00219-f005:**
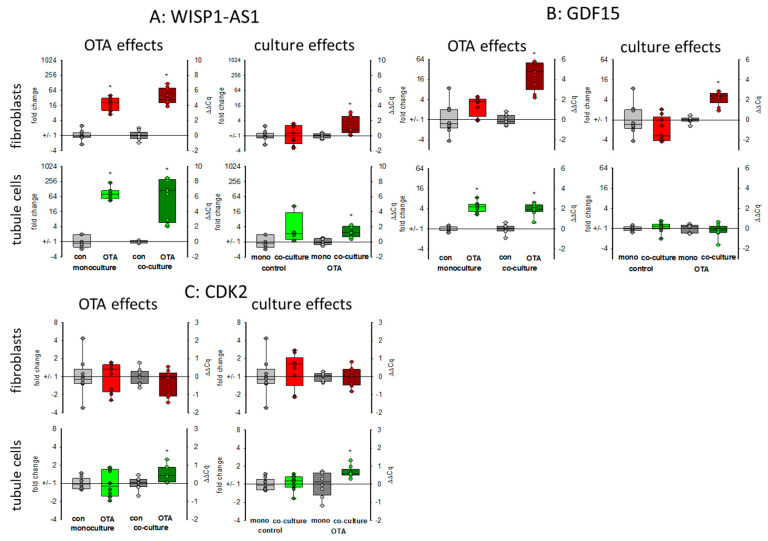
Effects of OTA and/or culture conditions on expression of RNA coding for *WISP1-AS1* (**A**), *GDF15* (**B**), and *CDK2* (**C**). Cells were cultivated either in mono- or in co-culture and exposed to 100 nM OTA for 48 h. *n* = 3, *n* = 4–9. * indicates a *p* < 0.05 to non-OTA-exposed cells (comparing the OTA effects, resp. left side) or to cells in mono-culture (comparing culture effects, resp. right side).

**Figure 6 toxins-13-00219-f006:**
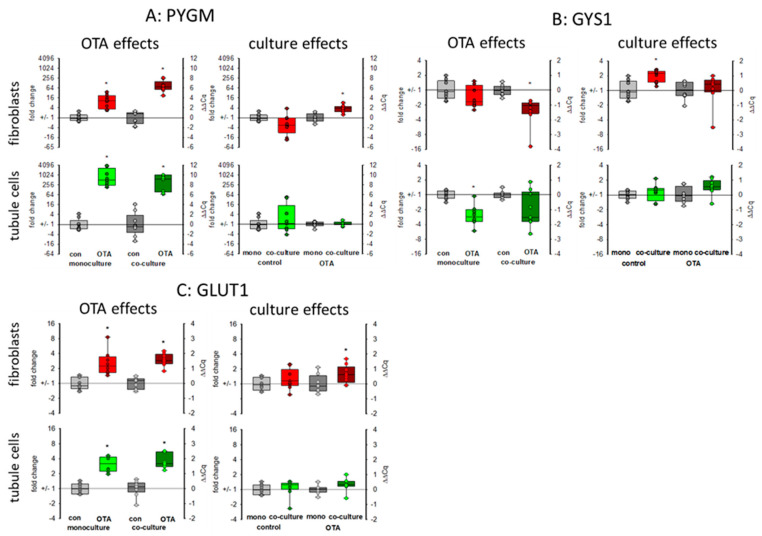
Effects of OTA and/or culture conditions on expression of RNA coding for *PYGM* (**A**), *GYS1* (**B**), and *GLUT1* (**C**). Cells were cultivated either in mono- or in co-culture and exposed to 100 nM OTA for 48 h. *n* = 3, *n* = 8–9. * indicates a *p* < 0.05 to non-OTA-exposed cells (comparing the OTA effects, resp. left side) or to cells in mono-culture (comparing culture effects, resp. right side).

**Figure 7 toxins-13-00219-f007:**
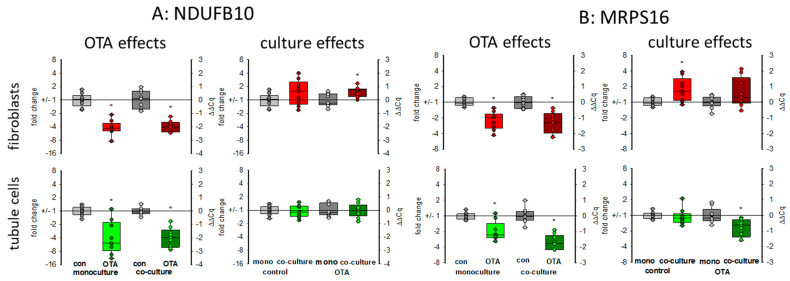
Effects of OTA and/or culture conditions on expression of RNA coding for *NDUFB10* (**A**) and *MRPS16* (**B**). Cells were cultivated either in mono- or in co-culture and exposed to 100 nM OTA for 24 h. *n* = 3, *n* = 8–9. * indicates a *p* < 0.05 to non-OTA-exposed cells (comparing the OTA effects, resp. left side) or to cells in mono-culture (comparing culture effects, resp. right side).

**Table 1 toxins-13-00219-t001:** Primer Sequences (in 5′–3′) used in Real-time-PCR.

Gene Name	Forward	Reverse	Fragment Length
*CDKN1A*	ACTGTCTTGTACCCTTGTGC	CTCTTGGAGAAGATCAGCCG	144
*CDK2*	ATTCATGGATGCCTCTGCTC	TTTAAGGTCTCGGTGGAGGA	122
*EEF2*	GGAGTCGGGAGAGCATATCA	GGGTCAGATTTCTTGATGGG	108
*FN*	CCATAAAGGGCAACCAAGAG	AAACCAATTCTTGGAGCAGG	142
*GDF15*	CTCCAGATTCCGAGAGTTGC	CACTTCTGGCGTGAGTATCC	130
*GYS1*	TTCTACAACAACCTGGAG	CTGAGCAGATAGTTGAGC	404
*NDUFB10*	ATGATGAAAGCGTTCGACCT	TTGCACTCAGTGATGTCTGG	137
*MRPS16*	AGAAAAACTCGTTGCCCTCA	AGCAAGACCCAGAAGCTTTT	97
*PYGM*	TCAATGTCGGTGGCTACATC	CACCACGAAATACTCCTGCT	131
*RPS17*	TCAGCCTTGGATCAGGAGAT	CATCCCAACTGTAGGCTGAG	114
*SLC2A1* (GLUT1)	ACACTGGAGTCATCAATGCC	ACACTGGAGTCATCAATGCC	148
*VIM*	ATTGCAGGAGGAGATGCTTC	TTCCACTTTGCGTTCAAGGT	112

**Table 2 toxins-13-00219-t002:** Antibodies used in Western Blot Experiments.

Antibody Against	Source	Dilution
CDKN1A/p21	Cell Signaling	0.7361111
COX2	Abcam	0.3888889
Fibronectin	Rockland	0.7361111
VIM	Cell Signaling	0.7361111
Beta-Actin	Cell Signaling	0.7361111
Mouse antibody (2nd antibody)	Licor	1:40,000
Rabbit Antibody (2nd antibody)	Licor	1:40,000

## Data Availability

The data presented in this study are available in the Appendix A.

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
