# Peer review of "The Impact of the Nephrotoxin Ochratoxin A on Human Renal Cells Studied by a Novel Co-Culture Model Is Influenced by the Presence of Fibroblasts"

_toxins, 2021, doi:10.3390/toxins13030219_

Round 1

Reviewer 1 Report

Although several in vivo works have already demonstrated the mechanisms of  OTA action at the renal level, the co-culture model proposed here seems very interesting.

These modifications of the following points are required:

line 29-30: "several studies have been performed" but none are indicated. Please add some references (doi: 10.1016 / j.tox.2017.03.009. -doi: 10.1002 / jcp.26753. Doi: 10.3390 / antiox9040332.)
 line 40-44: primary cell lines may have problems with data repeatability, please, motivate the choice of this cell culture.
line 125: motivate why you have studied only p21
line 140: who is COX? please write in full-text. Furthermore, these data seem to contradict already pubblished data . Please re-write you data more clearly and motivate them.
line 160: fibrosis exists in case of renal damage, justify the data of OTA-effect was lower in co-culture
line 293: standardize the references to the text

Author Response

We thank the reviewers for their critical suggestions. We corrected the manuscript according to their suggestions as listed below:

  1. line 29-30: "several studies have been performed" but none are indicated. Please add some references (doi: 10.1016 / j.tox.2017.03.009. -doi: 10.1002 / jcp.26753. Doi: 10.3390 / antiox9040332.)

We included the mentioned references in the introduction section.

  1. line 40-44: primary cell lines may have problems with data repeatability, please, motivate the choice of this cell culture

In the study, we did not use primary cells. HK2 and fibroblast cells are cell lines. Therefore, the mentioned problems with repeatability are excluded (at least those coming from primary cells)

  1. line 125: motivate why you have studied only p21

There are a lot of proteins related to cell cycle. We chose to study two of them: p21 and CDK2, which is shown in section 2.3.3, because they have been shown to be regulated by OTA (see reference 21, Dubourg et al.) to study possible additional effects due to mono- or co-culture conditions.

  1. line 140: who is COX? please write in full-text. Furthermore, these data seem to contradict already pubblished data . Please re-write you data more clearly and motivate them.

COX2 is cyclooxygenase 2. We added this. We can only show the results of our experiments. In studies with rat cells, the response of fibroblasts was similar to the results shown here. We added this in the discussion section (lit #20)

  1. line 160: fibrosis exists in case of renal damage, justify the data of OTA-effect was lower in co-culture

We show the results of our experiments. The observation that the results may be contradictory is discussed in the discussion section

  1. line 293: standardize the references to the text

We carefully controlled the references

Reviewer 2 Report

The manuscript entitled ‘The Impact of the Nephrotoxin Ochratoxin A on Human Renal 2 Cells Studied by a Novel Co-Culture Model Is Influenced by 3 the Presence of Fibroblasts’, the authors have investigated the effect of Ochratoxin A on human renal cells by a co-culture model. The manuscript has merit and may be considered for publication after a major revision. Authors need to address the following comments.

  1. In the introduction section authors should include the reported amounts of Ochratoxin found in human food. They should consider reference 15 for better flow and information that must be included in the introduction PMID: 31415839

DOI: 10.1016/j.bbamcr.2019.118528

  1. In section 2.1 authors should include a cartoon showing how they have used the co-culture technique; something similar depiction can me made such as used in reference 15, this will be helpful for other researchers to reproduce or to use this technique with ease.

  1. In section 2.1 line 97 authors have mentioned ‘LDH release as a measure for necrosis’, according to the literature this assay is used to measure cell death (apoptosis and necrosis) not specifically necrosis.

  1. In section 2.1 authors mentioned (line 102) ‘In fibroblasts, OTA exposure led to a small increase in protein content whereas in tubule cells OTA led to a slight decrease of protein content’, What does author mean by the protein content, what information did they get by protein content ? Why and how this assay was performed? Use a better quality pictures in figure 1, X and Y-axis markings are not clear.

What is % resp. control? Please elaborate in the figure legend section and how many times the experiment was repeated. Please write very briefly the findings in the figure legends.

5 Authors should consider including a proper apoptosis assay (Annexin V/ PI staining- by Flow cytometry) to have clear picture of what percentage of cells undergone apoptosis and necrosis.

  1. In figure 1, there are different panels (diff experiments) , authors should mark them 1A, 1B, 1C… and same should be mentioned in the result section while explaining the result and findings.

  1. In section 2.2.1. Authors have analyzed CDKN1A/p21 expression levels; p21 is a cell cycle dependent kinase inhibitor, which is up regulated in case of cell cycle arrest, did authors check for cell cycle analysis? If not, they may consider Propidium Iodide (PI) staining and analyze cell cycle distribution in different phases of the cell cycle.

  1. The quality of figures is not good, please replace with better quality picture with clearly visible x and y-axis. Do point 4 for all figures, legends, and result section.

Minor comment- Authors need to improve the English language and the flow of the entire manuscript

Author Response

We thank the reviewers for their critical suggestions. We corrected the manuscript according to their suggestions as listed below:

1 In the introduction section authors should include the reported amounts of Ochratoxin found in human food. They should consider reference 15 for better flow and information that must be included in the introduction PMID: 31415839

DOI: 10.1016/j.bbamcr.2019.118528

We included a reference for the OTA concentration in food and also added the reference additionally a bit earlier. former ref 15 was originally placed one sentence later.

  1. In section 2.1 authors should include a cartoon showing how they have used the co-culture technique; something similar depiction can me made such as used in reference 15, this will be helpful for other researchers to reproduce or to use this technique with ease.

  1. In the graphical abstract a scheme of the co-culture is shown. We refer to this scheme in this section.

  1. In section 2.1 line 97 authors have mentioned ‘LDH release as a measure for necrosis’, according to the literature this assay is used to measure cell death (apoptosis and necrosis) not specifically necrosis.

  1. To our best knowledge, a hallmark of apoptosis is that during apoptosis the cells die but the integrity of the membrane remains intact thus excluding the release of intracellular components which would lead to inflammatory events. During necrosis, the cell membranes become leaky and e.g. LDH is released. Therefore, LDH release is a measure for necrosis only and cannot be a measure for apoptosis. These assays (LDH release and caspase-3) have been used widely to study apoptosis and necrosis.

4    In section 2.1 authors mentioned (line 102) ‘In fibroblasts, OTA exposure led to a small increase in protein content whereas in tubule cells OTA led to a slight decrease of protein content’, What does author mean by the protein content, what information did they get by protein content ? Why and how this assay was performed? Use a better quality pictures in figure 1, X and Y-axis markings are not clear.

  1. In figure 1, there are different panels (diff experiments) , authors should mark them 1A, 1B, 1C… and same should be mentioned in the result section while explaining the result and findings.

What is % resp. control? Please elaborate in the figure legend section and how many times the experiment was repeated. Please write very briefly the findings in the figure legends

4 and 6: We explained the meaning of measuring protein content (a crude and additional overall measure for cell status). We added A, B and C for a better understanding in the pictures and in the text. Also in the legends we added an explanation for what is resp control (depending on what was compared: either the non-OTA-exposed cells served as controls or the cells im mono-culture). The number of experiments is added (N = ). Would be difficult to very briefly summarize the findings as they are quite complex.

  1. Authors should consider including a proper apoptosis assay (Annexin V/ PI staining- by Flow cytometry) to have clear picture of what percentage of cells undergone apoptosis and necrosis
  2. The use of additional assays to get a further insight is a good idea. But we think, that the caspase-3 assay already gives a good idea about the amount of apoptosis. To our experience, whenever we used additionally another assay, the results were very well comparable. Therefore, we restricted to the caspase-3 activity which is to our opinion a good indicator of apoptosis.

  1. 7. In section 2.2.1. Authors have analyzed CDKN1A/p21 expression levels; p21 is a cell cycle dependent kinase inhibitor, which is up regulated in case of cell cycle arrest, did authors check for cell cycle analysis? If not, they may consider Propidium Iodide (PI) staining and analyze cell cycle distribution in different phases of the cell cycle.

  1. It is certainly a good idea to analyze cell cycle. Partially, these experiments using tubule cells alone have been done already as described in ref. 21. In the discussion section we added this as a possibility for further studies (“Cell cycle studies may be added to get a further insight into the impact on cell cycle.”).

  1. The quality of figures is not good, please replace with better quality picture with clearly visible x and y-axis. Do point 4 for all figures, legends, and result section.

We improved the figures and mention the resp. pictures in the text according to their appearance.

Minor comment- Authors need to improve the English language and the flow of the entire manuscript.

A native speaker has had a look on the manuscript.

Round 2

Reviewer 1 Report

The paper is ready to pubblication

Reviewer 2 Report

The authors have addressed the concerns in the revision. The manuscript may be considered for publications.